# Ipilimumab, Pembrolizumab, or Nivolumab in Combination with BBI608 in Patients with Advanced Cancers Treated at MD Anderson Cancer Center

**DOI:** 10.3390/cancers14051330

**Published:** 2022-03-04

**Authors:** Henry Hiep Vo, Carrie Cartwright, I-Wen Song, Daniel D. Karp, Graciela M. Nogueras Gonzalez, Yuran Xie, Michael Karol, Matthew Hitron, David Vining, Apostolia-Maria Tsimberidou

**Affiliations:** 1Department of Investigational Cancer Therapeutics, The University of Texas MD Anderson Cancer Center, 1515 Holcombe Blvd., Houston, TX 77030, USA; htvo2@mdanderson.org (H.H.V.); cacartwr@mdanderson.org (C.C.); isong1@mdanderson.org (I.-W.S.); dkarp@mdanderson.org (D.D.K.); 2Department of Biostatistics, The University of Texas MD Anderson Cancer Center, 1515 Holcombe Blvd., Houston, TX 77030, USA; gnoguera@mdanderson.org; 3Sumitomo Dainippon Pharma Oncology, 640 Memorial Drive, Cambridge, MA 02139, USA; yuran.xie@sdponcology.com (Y.X.); michael.karol@sdponcology.com (M.K.); matthew.hitron@sdponcology.com (M.H.); 4Department of Abdominal Imaging, The University of Texas MD Anderson Cancer Center, 1515 Holcombe Blvd., Houston, TX 77030, USA; dvining@mdanderson.org

**Keywords:** advanced cancer, clinical trial, BBI608, targeted therapy, immunotherapy, checkpoint inhibitor

## Abstract

**Simple Summary:**

BBI608 is an investigational reactive oxygen species generator that affects several molecular and oncogenic pathways, including the STAT3 pathway, and may overcome resistance to immune checkpoint inhibitors. We investigated BBI608 combined with immunotherapy (ipilimumab, pembrolizumab, or nivolumab) in patients with advanced cancer. Treatment was well tolerated overall. Only 2 of 12 patients had Grade 3 diarrhea. Five patients treated with BBI608/nivolumab had prolonged disease stabilization lasting for 12.1, 10.1, 8.0, 7.7 and 7.4 months. Four patients had prolonged overall survival (53.0, 48.7, 51.9 and 48.2 months). Prospective studies of BBI608 are warranted.

**Abstract:**

**Background**: BBI608 is an investigational reactive oxygen species generator that affects several molecular pathways. We investigated BBI608 combined with immune checkpoint inhibitors in patients with advanced cancers. **Methods**: BBI608 (orally twice daily) was combined with ipilimumab (3 mg/kg IV every 3 weeks); pembrolizumab (2 mg/kg IV every 3 weeks); or nivolumab (3 mg/kg IV every 4 weeks). We assessed the safety, antitumor activity and the pharmacokinetic profile of BBI combined with immunotherapy. **Results:** From 1/2017 to 3/2017, 12 patients were treated (median age, 54 years; range, 31–78; 6 men). Treatment was overall well tolerated. No dose-limiting toxicity was observed. The most common adverse events were diarrhea (5 patients: grade (G)1–2, *n* = 3; G3, *n* = 2) and nausea (4 patients, all G1). Prolonged disease stabilization was noted in five patients treated with BBI608/nivolumab lasting for 12.1, 10.1, 8.0, 7.7 and 7.4 months. The median progression-free survival was 2.73 months. The median overall survival was 7.56 months. Four patients had prolonged overall survival (53.0, 48.7, 51.9 and 48.2 months). **Conclusions**: Checkpoint inhibitors combined with BBI608 were well tolerated. Several patients had prolonged disease stabilization and overall survival. Prospective studies to elucidate the mechanisms of response and resistance to BBI608 are warranted.

## 1. Introduction

BBI608 (napabucasin, Boston Biomedical Incorporated, Cambridge, MA, USA) is an investigational, orally administered reactive oxygen species (ROS) generator bioactivated by the intracellular antioxidant nicotinamide adenine dinucleotide phosphate (NAD[P]H): quinone oxidoreductase 1 [1,2]. Production of ROS may affect several molecular pathways, including the signal transducer and activator of transcription 3 (STAT3) pathway, and if ROS is produced in sufficient amounts, it can overwhelm anti-oxidant defenses and cause cell damage and death. STAT3 is an oncogene that mediates gene expression and metabolic regulations in various human solid tumors and hematologic malignancies and plays an important role in the growth, proliferation, survival, maintenance, and self-renewal of cancer stem cells (CSCs) [3,4,5,6,7], which are also capable of differentiating and maintaining tumor heterogeneity. Although in vitro data demonstrated that napabucasin can inhibit CSCs and may induce apoptosis in both CSCs and heterogeneous cancer cells, anti-CSC activity of napabucasin has not been demonstrated in patient-derived tumors.

Though the in vitro anti-cancer activity of napabucasin does not depend on STAT3 inhibition, it is possible that inhibition of this pathway has in vivo effects. Such effects are theoretical and based on the understanding that STAT increases the expression of oncogenic genes such as (sex-determining region Y)-box 2 (Sox2), c-Myc oncoprotein (c-Myc), homeobox transcription factor Nanog (Nanog), and β-catenin and confers immune evasion by directly activating transcription of programmed cell death protein 1 (PD-1), programmed death-ligand 1 (PD-L1), and programmed death-ligand 2 (PD-L2) [8,9,10,11]. The STAT3-Janus kinase 3 (JAK3) pathway also mediates cytotoxic T-lymphocyte-associated protein 4 (CTLA4)-induced immune suppression in malignant B cells [12] and enhances IL-10-stimulated CTLA4 expression in regulatory T-cells [13]. In vitro preclinical experiments with napabucasin have shown that it inhibits Nanog, Axl receptor tyrosine kinase (Axl), Sox-2, kruppel-like factor 4 (Klf4), survivin, c-Myc, BMI1 proto-oncogene, polycomb ring finger (Bmi-1), and β-catenin [14]. These effects are presumed to be secondary to the generation of ROS, and it is unclear whether they can enhance the activity of immune checkpoint inhibitors [14].

Immune checkpoint inhibitors offer an innovative and promising approach for the treatment of patients with cancer [15]. Checkpoint blockade immunotherapy utilizes different monoclonal antibodies to directly target immune evasion mechanisms including PD-1/PD-L1 and CTLA-4 pathways, resulting in the elimination of cancer cells. Ipilimumab is a checkpoint inhibitor targeting CTLA-4 that was first approved by the U.S. Food and Drug Administration (FDA) in 2011 [16], while nivolumab and pembrolizumab, which both target PD-1, were first approved by the FDA in 2014 [17].

Despite the promising clinical outcomes of checkpoint inhibitors in various tumor types, the response rate is only approximately 20%, and a significant proportion of initial responders develop resistance eventually. One key mechanism underlying cancer cell resistance to immunotherapy is associated with the ability of CSCs to avoid immune detection and elimination through activation of immune evasion pathways. Moreover, checkpoint inhibitor-related serious immune-related adverse events (irAEs) have been reported to account for over 60% of the total irAEs associated with immunotherapy [18,19,20]. Additionally, in a subset of patients with advanced cancer treated with checkpoint inhibitors, accelerated disease progression occurs, along with an increased tumor growth rate [21,22,23].

Given the activity of ROS against STAT3 and the strong scientific rationale for STAT3 being a critical driver for cancer stemness and immune evasion, an investigation of BBI608 in combination with immune checkpoint inhibitors was imperative. Therefore, we conducted a first-in-human study of BBI608 in combination with the immune checkpoint inhibitors ipilimumab, pembrolizumab, or nivolumab in adult patients with advanced cancers. Notably, when the study was initiated, BBI608 was thought to be a STAT3 and a cancer stemness inhibitor [14,24,25,26]. The rationale for combining BBI608 with checkpoint inhibitors was based on the hypothesis that inhibition of cancer stem cells and STAT3, a mechanism of tumor evasion from immune surveillance and resistance to immunotherapy, would overcome tumor resistance to immunotherapy. However, data emerging from subsequent studies have demonstrated that BBI608 is not a direct STAT3 inhibitor but a ROS generator. We assessed the safety tolerability and the preliminary anti-tumor activity of BBI608 administered orally, daily, in combination with ipilimumab, or nivolumab, or pembrolizumab in patients with advanced cancers. We also determined the pharmacokinetic profile of BBI608 when administered in combination with ipilimumab, or nivolumab, or pembrolizumab.

## 2. Patients and Methods

Patients were recruited from The University of Texas MD Anderson Cancer Center. Eligible patients were ≥18 years old with a confirmed diagnosis of advanced malignancy that was metastatic, unresectable, or recurrent and for which treatment with ipilimumab, nivolumab, or pembrolizumab was a reasonable therapeutic option in the opinion of the investigator. Other eligibility criteria included measurable or evaluable disease by Response Evaluation Criteria in Solid Tumors (RECIST) 1.1; European Cooperative Oncology Group (ECOG) performance status of 0–1; adequate bone marrow (absolute neutrophil count ≥1.5 × 10^9^/L, hemoglobin ≥9 g/dL, platelets ≥100 × 10^9^/L), hepatic (total bilirubin level ≤1.5 mg/dL, unless the patient had known Gilbert’s disease, and alanine transaminase and aspartate transaminase levels ≤2.5 times the upper limit of normal (ULN) without liver metastases or ≤3.5 times the ULN with liver involvement), and renal (serum creatinine ≤1.5 × ULN or creatinine clearance ≥60 mL/min estimated using the Cockcroft–Gault formula) function; and predicted life expectancy of ≥12 weeks. Patients with brain metastases were eligible if the metastases had been stable (treated and asymptomatic) for at least 3 weeks and required no corticosteroid therapy. Men and women of childbearing potential were required to use adequate contraception prior to study entry and for the duration of study participation.

Patients were excluded from this study if they had adenocarcinoma of unknown primary; had received anti-cancer chemotherapy, radiotherapy, immunotherapy, or investigational agents within the past 7 days; or had undergone a surgical procedure <4 weeks before the first dose of BBI608. All adverse events (AEs) from prior therapy had to be resolved to grade 1 prior to entering the study. Other exclusion criteria included leptomeningeal metastases and pregnancy or breastfeeding. Patients were excluded if they were unable to swallow BBI608 capsules, had significant gastrointestinal (GI) disorders that would impair drug absorption, or had an active autoimmune disease requiring immunosuppression, except for isolated vitiligo, controlled hypoadrenalism or hypopituitarism, and a history of Grave’s disease in euthyroid patients. Patients with concurrent active malignancy, interstitial lung disease, known hypersensitivity to a component of protocol therapy, or uncontrolled concurrent illness were also excluded.

All patients signed an informed consent form before treatment, according to institutional policy, stating that they were aware of the experimental nature of the clinical trial. The MD Anderson Cancer Center institutional review board approved the study protocol. The study was registered at www.clinicaltrials.gov (accessed on 1 February 2022) (NCT02467361). This was a multicenter study. For logistical reasons, we report only on the patients treated at MD Anderson Cancer Center (Houston, TX, USA). The study was sponsored by Boston Biomedical, and it was conducted in accordance with ICH Good Clinical Practice and the Declaration of Helsinki.

### 2.1. Treatment

The study design is shown in Table 1 and Figure 1. The first dose level of BBI608 was 240 mg orally (*per os*, PO) twice daily (BID), which was approximately 24% of the maximally administered total daily dose (2000 mg) and 50% of the recommended phase II dose (RP2D; 480 mg orally BID, 960 mg total daily). The immune checkpoint antibodies were administered at the FDA-approved doses at the time of the study. For each patient, the appropriate immunotherapeutic agent was determined by the treating physician. BBI608 was administered in combination with 3 mg/kg ipilimumab administered intravenously over 90 min every 21 days for a total of four doses, in combination with 3 mg/kg nivolumab administered intravenously over 60 min every 14 days, or in combination with 2 mg/kg pembrolizumab administered intravenously over 30 min every 21 days. Each dose of BBI608 was taken with fluids, either 1 h before a meal or 2 h after a meal. On days when an immunotherapeutic agent was being infused, BBI608 was administered approximately 1 h prior to ipilimumab, nivolumab, or pembrolizumab.

**Premedication**: To prevent diarrhea and abdominal cramping, patients received 4 mg of loperamide or 5 mg of diphenoxylate/atropine twice daily 24 h prior to the first dose of BBI608 on cycle 1, day 1. Anti-nausea medications (ondansetron or other 5HT3 receptor inhibitors) were also administered per standard procedures.

**Supportive treatment**: Patients who developed diarrhea and abdominal cramping received dicyclomine, diphenoxylate/atropine with or without loperamide, systemic opioids, hyoscine, or budesonide. Patients who developed nausea/vomiting received 5HT3 receptor inhibitors as first-line agents, and dexamethasone was added if nausea/vomiting persisted. Antihistamines, benzodiazepines, proton pump inhibitors/H2 antagonists, dopamine antagonists, and cannabinoids could also be used.

**Patient monitoring**: Patients were monitored closely, with weekly visits through the first cycle of combination protocol therapy. Starting with cycle 2, all patients were evaluated on the first day of each cycle, and patients in the nivolumab arm were also evaluated on day 15 of each cycle.

### 2.2. Study Design

The starting dose levels and dose modifications for BBI608 administration are described in Appendix A. The starting dose level was administered to an initial cohort of six patients. Additional patients were enrolled according to the number of patients from the initial dose-level cohort who experienced dose-limiting toxicity (DLT). The RP2D for a given study cohort was the dose level at which ≤one of six patients experienced DLT (Appendix A).

Treatment was discontinued when any of the following criteria were met: disease progression, unacceptable adverse event(s), consent withdrawal, intercurrent illness unrelated to protocol therapy or cancer, loss to follow-up/death, or noncompliance. Discontinuing BBI608 or immunotherapy and continuing with the remaining agent as monotherapy was allowed after discussion with the medical monitor for the sponsor. BBI608 could be continued beyond radiologic progression of disease, provided there was no clinical deterioration.

Ipilimumab was administered for a maximum of four doses, though BBI608 could be continued beyond the end of ipilimumab administration. Pembrolizumab and nivolumab were administered until disease progression or unacceptable toxicity. For patients enrolled in the pharmacokinetic cohort, the immunotherapy infusion began precisely 1 h following the morning dose of BBI608 (±15 min).

We also performed tumor molecular profiling according to our procedures. Patient tumor samples underwent next-generation sequencing (NGS) using solid tumor genomic assay at MD Anderson or in a Clinical Laboratory Improvement Amendments (CLIA)-certified laboratory. Briefly, DNA was extracted from formalin-fixed, paraffin-embedded tumor tissue samples and subjected to NGS that detects substitutions, insertion and deletion alterations, copy number alterations, and gene rearrangements, as previously described [27,28,29,30].

### 2.3. Pharmacokinetic Studies

Blood samples were to be collected from patients for comprehensive pharmacokinetic (PK) characterization of BBI608 at 0, 0.5, 1, 2, 3, 4, 5, 6, 7, 8, 10 and 12 h after infusion of the immune checkpoint inhibitor on cycle 1, day 1 and on cycle 2, day 1 (specifically, on a day in which both BBI608 and immunotherapy were administered). The concentration (ng/mL) of BBI608 was measured in patients’ plasma using liquid chromatography with tandem mass spectrometry (LC-MS/MS).

### 2.4. Study Endpoints and Statistical Analysis

Toxicities were assessed using the National Cancer Institute Common Terminology Criteria for Adverse Events (NCI CTCAE), version 4.0. DLTs were assessed during the initial first 6 weeks of treatment and were defined as follows: (1) grade 4 hematologic toxicity, including grade 3 thrombocytopenia with significant bleeding requiring transfusion or hospitalization and febrile neutropenia (absolute neutrophil count <1000/mm^3^ and temperature >38.3 °C or sustained temperature of ≥38 °C for >1 h); (2) alanine aminotransferase /aspartate aminotransferase >3 times the ULN with bilirubin >2 times the ULN without another explanation; (3) grade 3–4 non-hematological toxicity, with the exception of grade 3 nausea/vomiting or grade 4 vomiting that resolved within 72 h, grade 3–4 diarrhea that resolved within 72 h, grade 3–4 not clinically significant laboratory abnormalities, grade 3 fatigue lasting <5 days, and controlled grade 3 hypertension; (4) grade 3 immune-related adverse reactions that persisted for >7 days despite corticosteroid treatment; (5) grade ≥2 pneumonitis not improving with corticosteroids; and (6) grade ≥2 uveitis. Alopecia was not a DLT. Patients had to receive ≥75% of the assigned dose of protocol therapy during the DLT evaluation period to be eligible for DLT determination.

Patients were allowed to remain on the study until progression of disease, death, withdrawal of consent, or unacceptable toxicity occurred.

Descriptive statistics and frequency tables were used to summarize the patients’ characteristics and safety data. Response to treatment was assessed and estimated according to the immune-related Response Evaluation Criteria in Solid Tumors (irRECIST) along with 95% confidence intervals [31,32]. Progression-free survival (PFS) was measured from the start of treatment on protocol until progression or death due to any cause, whichever occurred first. Overall survival (OS) was measured from the start of treatment on protocol until death from any cause or until last follow-up. Survival endpoints were estimated using the Kaplan–Meier method. A paired time-to-event analysis was used for comparing PFS between treatment with BBI608 plus immunotherapy and patients’ prior therapy [33]; the Andersen–Gill Cox model was fit to determine differences in PFS using multiple failure-time data [34]. A *p*-value < 0.05 was considered statistically significant. Statistical analysis was performed using Stata/SE, version 16.1, statistical software (Stata Corp., LP, College Station, TX, USA).

## 3. Results

### 3.1. Patients

Overall, 104 patients were treated on protocol in all participating institutions. From January 2017 to March 2017, 16 patients were screened, and 12 patients were treated at MD Anderson. The remaining four patients were not treated because of consent withdrawal (*n* = 1), selection of another study (*n* = 1), insurance issues (*n* = 1), and identification of bronchiolitis obliterans with organizing pneumonia on screening (*n* = 1). Table 2 summarizes the baseline characteristics of the 12 treated patients. The median age was 54 years (range, 31–78 years), and six (50%) patients were men. Six (50%) patients had ≥2 metastatic sites (range, 1–6) and three patients (25%) had hepatic metastases. The median number of prior therapies was three (range, 1–5). Patient tumor molecular profiles are listed in Appendix A.

### 3.2. Treatment

Eight (67%) of 12 patients were treated on the nivolumab cohort, three in the pembrolizumab cohort, and one in the ipilimumab cohort (Figure 1). The median number of treatment cycles was seven (range, <1–13 cycles). Patients remained on treatment for a median of 3.3 months (range, 0.1–12.1 months). Nine (75%) patients completed >75% of the assigned doses of protocol therapy. Three patients (patients 02, 09, and 11) required dose reduction of napabucasin because of gastrointestinal symptoms (see Safety section). Patient 02, who was treated with BBI608 at the 240 mg PO BID dose level in combination with pembrolizumab, required dose reduction (cycle 1, day 14) to 80 mg PO BID and was taken off protocol on cycle 1, day 21 owing to intolerance of BBI608. Patient 09, who was treated with BBI608 at the 240 mg PO BID dose level in combination with nivolumab, required a dose decrease to 80 mg PO daily (cycle 1, day 35) because of worsening preexisting peripheral neuropathy and continued treatment for 7 cycles. Patient 11, who was treated with BBI608 at the 480 mg PO BID dose level in combination with pembrolizumab, required a dose reduction to 80 mg PO BID (cycle 1, day 13) and continued treatment for 11 cycles.

### 3.3. Safety

Treatment was overall well tolerated in this heavily pretreated patient population. Treatment-emergent adverse events (TEAEs) are summarized in Table 3. TEAEs by patient are listed in Appendix A. The most common adverse events were gastrointestinal and included grade 1–3 diarrhea, nausea, and abdominal pain/cramping. Four patients experienced grade 3–4 irAEs, all gastrointestinal-related (Appendix A).

Patients 03, 04 and 05 received oral budesonide for treatment of diarrhea and abdominal pain. Two patients (01 and 11) developed grade 3 colitis. Patient 01, a female in her late 60s with colon cancer, experienced abdominal pain on cycle 1, day 5, and a computed tomography (CT) scan of the abdomen demonstrated colitis. She was hospitalized for 2 days and treated with steroids. She was taken off study on cycle 1, day 5 owing to radiologic evidence of new brain metastasis, indicating disease progression. Patient 11 was a female in her late 30s with adenoid cystic adenoma of the parotid gland who experienced diarrhea and hematochezia on cycle 1, day 2. The study treatment was suspended, and the patient was hospitalized. A CT scan of the abdomen demonstrated diffuse colorectal wall thickening consistent with infectious colitis. Flexible sigmoidoscopy demonstrated erythema and a solitary ulcer. She was treated with metronidazole for 2 weeks. After discussion with the sponsor, the dose of BBI608 was decreased from 480 mg PO BID to 80 mg PO BID starting on cycle 1, day 13. The patient completed 11 cycles of treatment until she developed progressive disease.

One patient (02) experienced BBI608 intolerance and progressive disease (PD) and discontinued treatment. He was a male in his 70s with metastatic mesothelioma who required hospitalization for nausea, abdominal pain, cramping, and diarrhea without blood or mucus on cycle 1, day 3. He resumed BBI608 on cycle 1, day 14, with a dose decrease from 240 mg PO BID to 80 mg PO BID. On cycle 1, day 21, the patient was hospitalized for worsening shortness of breath and leukocytosis attributed to PD, as evidenced by CT imaging.

A male in his late 60s with metastatic pancreatic cancer (patient 06) developed grade 3 anemia, fatigue, and abdominal pain starting on cycle 1, day 11. He was hospitalized twice, and on cycle 1, day 26, treatment was discontinued because of weakness, hypoxemia, and failure to thrive attributed to PD.

A female in her late 60s with metastatic lung cancer (patient 07) required hospitalization for pulmonary embolism on cycle 2, day 19. CT imaging also demonstrated new liver metastases, and treatment was discontinued owing to PD.

A male in his early 30s with carcinoma ex-pleomorphic adenoma of the right parotid gland metastatic to the lungs (patient 10) required hospitalization for grade 3 hemoptysis on cycle 8, day 4. The study drug was placed on hold and the patient underwent bronchoscopy for thermal therapy of the left main stem bronchus. His hemoptysis resolved, but 5 days later the patient was re-admitted to the hospital for bronchitis and treated successfully with antibiotics. He completed nine cycles of treatment before he developed PD.

### 3.4. Clinical Outcomes

Clinical outcomes of the 12 patients are summarized in Table 4. Prolonged disease stabilization (≥6 months) was noted in five patients (04, 09, 10, 11, and 12) and lasted for 12.1, 7.4, 8.0, 10.1 and 7.7 months, respectively. No objective response was noted. Response and clinical events are illustrated in a swimmer plot (Figure 2), and changes in tumor measurement from baseline over time are illustrated in a spider plot (Figure 3A). The median PFS was 2.73 months (95% CI, 0.66–8.05 months) (Figure 3B).

The median OS was 7.56 months (95% CI, 1.22 months-not estimable) (Figure 3C and Table 4). Eight patients died from PD. Four patients (04, 09, 11 and 12) treated with BBI608 and nivolumab had prolonged OS (>4 years), lasting for 53.0+, 48.7+, 51.9+ and 48.2+ months, respectively. These four patients were still alive at the time of the analysis (August 2021). Therapies used after the discontinuation of protocol therapy are shown in Table 4.

A woman in her early 50s with metastatic adenoid cystic carcinoma of the Bartholin gland (patient 04) remained on study for 13 cycles (BBI608, 240 PO BID combined with nivolumab) and had SD for 12.1 months. She was subsequently treated with investigational therapies (Table 4) and was still alive at the time of analysis (OS, 53.0+ months). Molecular profiling of her tumor demonstrated the following genetic mutations: CREBBP Y1450 C, MLL2 Q1949 * and PIK3R1 Q329 (Appendix A). PD-L1 expression of her tumor was 0%, as assessed using tumor cell staining (membranous) and tumor-associated immune cell staining.

A woman in her late 30s with adenoid cystic carcinoma of the parotid gland (patient 11) remained on study for 11 cycles (treated with BBI608, 480 PO BID and nivolumab) and had SD for 10.1 months; her OS duration was 51.9+ months. Her tumor molecular profile performed 2.5 years after completion of treatment with BBI608 demonstrated MYB proto-oncogene (MYB) chromosomal rearrangement, BCL6 corepressor (BCOR) P698fs, and lysine demethylase 6A (KDM6A) R1415 *. Her tumor PD-L1 expression scores assessed using tumor cell staining (membranous) and tumor-associated immune cell staining were <1% and 1%, respectively (Appendix A).

### 3.5. PFS with BBI608 and Immunotherapy Compared with PFS of Previous Systemic Therapy

Ten of twelve patients had received prior systemic therapy. Their median PFS with prior systemic therapy was 4.08 months (95% CI, 1.78–9.04). Their median PFS with BBI608 and immunotherapy combined was 2.73 months (95% CI, 0.13–7.4). There was no statistical difference in PFS between the prior and current regimens (*p* = 0.43; Appendix A).

### 3.6. Pharmacokinetic Studies

The pharmacokinetic profiles of the 12 patients treated at MD Anderson are shown in Appendix A. Elimination half-life, C_max_, area under the curve (AUC), and other PK parameters were not calculated because we report on patients treated at MD Anderson only (the remaining patients on the study were treated at other institutions and their data are unavailable).

## 4. Discussion

Treatments for patients with advanced, metastatic solid tumors vary, and multiple clinical trials that include immune-oncology agents are being conducted. Investigational treatment with targeted therapy and immunotherapy offers promise to these patients with limited therapeutic options.

Here, we report a first-in-human phase I/II study of BBI608 in combination with the checkpoint inhibitors ipilimumab, nivolumab, or pembrolizumab in patients with various tumor types and treatment-refractory and/or metastatic disease. BBI608 was initially thought to inhibit STAT3 and CSCs, but it was later shown to generate ROS, which may subsequently target multiple oncogenic pathways [8,9,10,11,12,13,14].

Overall, treatment with BBI608 and immunotherapy was feasible and well tolerated. Nine (75%) patients completed ≥ 75% of the assigned doses of treatment. The safety profile in these combination cohorts was consistent with that of BBI608 monotherapy [26,35,36]. The most common treatment-related adverse events were from the gastrointestinal tract. Four (25%) of 12 patients developed grade 3 GI toxicity that was manageable following the protocol guidelines (Supportive Care section) and reversible.

The clinical outcomes of BBI608 combined with checkpoint inhibitors were encouraging in these heavily pretreated patients with advanced, metastatic cancer who had failed a median of three prior therapies. There was no treatment-related mortality. The duration of PFS and OS was prolonged in selected patients. Four patients with prolonged OS lasting up to 53 months were alive at the time of the analysis. Prolonged disease stabilization (≥6 months) was noted in five patients and lasted for up to 12 months (Table 4). Notably, these five patients were treated in the nivolumab cohort and had adenoid cystic carcinoma (*n* = 3), adenocarcinoma of the ovary (*n* = 1), and ex pleomorphic adenoma of the parotid (*n* = 1). Prolonged disease stabilization should be interpreted with caution, keeping in mind that three patients had adenoid cystic carcinoma and one patient had ex pleomorphic adenoma of the parotid. The course of these tumor types is typically very long and indolent, but it may become aggressive. BBI608 in combination with immunotherapy did not induce objective responses, which may be explained by our patients’ advanced, metastatic disease status, intrinsic resistance, additional molecular or compensatory pathways involved in carcinogenesis, or other mechanisms.

The median PFS was 2.73 months. In our analysis, no statistical difference in PFS was noted between the current study and treatment with prior systemic therapy. As shorter PFS is typically associated with subsequent therapies, BBI608 and immunotherapy may have a beneficial impact on PFS. These results have to be interpreted with caution, however, because the number of patients is too small to draw meaningful conclusions and the study was not randomized; therefore, the *p*-value is descriptive only and has no inferential interpretation (Appendix A).

A comparison of our results with published data of patients treated with other agents that also inhibit STAT3 is shown in Appendix A. Among these studies, the most commonly reported adverse events were diarrhea, nausea, and vomiting, and disease stabilization and objective responses were noted. In a phase I dose-escalation study of BBI608 in 14 Japanese patients with advanced solid tumors, the most common drug-related adverse events were GI disorders, and two patients with colorectal cancer had SD [26]. These data have to be placed in perspective, given the ethnic pharmacogenomic differences [37].

In a randomized phase III trial, patients with refractory advanced colorectal cancer were randomized (1:1 ratio) to receive BBI608 (*n* = 138) vs. placebo (*n* = 144) after stratification by performance status, kirsten rat sarcoma viral oncogene homolog (KRAS) status, prior vascular endothelial growth factor inhibitor treatment, and time from diagnosis of metastatic disease. No difference was found in median OS (primary endpoint) between the BBI608 and placebo arms (4.4 months vs. 4.8 months, respectively) (hazard ratio (HR) = 1.13, *p* = 0.34). In the BBI608 arm, the most common AEs (any grade) were treatment-related diarrhea (79%), nausea (51%), and anorexia (38%). The most common grade ≥3 treatment-related AEs were abdominal pain (BBI608, 4% vs. placebo, 3%), diarrhea (15% vs. 1%), fatigue (10% vs. 6%), and dehydration (4% vs. 1%) [38]. In phospho-STAT3-positive patients, OS was longer in the BBI608 arm than in the placebo arm (median OS, 5.1 months vs. 3.0 months; HR = 0.41, *p* = 0.0025). The results suggested that STAT3 might be an important target for the treatment of patients with colorectal cancer who have increased phospho-STAT3 expression [38].

The tolerability and clinical response observed in our combinations of BBI608 and checkpoint inhibitors are in line with those reported with the use of single-agent immunotherapy and/or BBI608 combined with checkpoint inhibitors in advanced solid tumors. A detailed comparison of our study with a previously published study of checkpoint inhibitors is shown in Appendix A. As the use of checkpoint inhibitors has been associated with prolonged or delayed response in selected patients, even after discontinuation of treatment, the prolonged OS may be associated with the use of these treatments [39,40]. Interestingly, our patients with prolonged PFS and OS were all treated with nivolumab. Other investigators have reported that nivolumab has a greater effect on OS and PFS compared to pembrolizumab [41]. However, our patient sample is too small to draw robust clinical conclusions.

A phase I study of pembrolizumab in 30 patients with advanced solid tumors demonstrated acceptable toxicity and encouraging antitumor activity in diverse tumor types [42]. The most common treatment-related AEs were fatigue, nausea, and pruritus. No grade 3 AEs were reported, and three patients discontinued therapy owing to treatment-related grade 2 fatigue, pneumonitis, and decreased weight [42].

In a phase I trial, nivolumab was well tolerated and induced antitumor activity in three (7.7%) of 39 patients with refractory solid tumors. One durable complete response (CR) and two partial responses (PRs) were noted. The most common treatment-related AEs included decreased CD4+ lymphocyte counts, lymphopenia, fatigue, and musculoskeletal toxicity. Grade 3 inflammatory colitis was observed in one patient with melanoma [43].

Literature reporting the safety and efficacy of ipilimumab monotherapy in patients with advanced, solid tumors is limited. The commonly observed AEs include fatigue, colitis/diarrhea, rash, and pruritus [44]. A phase I study of ipilimumab in 33 pediatric patients with advanced solid tumors demonstrated acceptable toxicity and encouraging antitumor activity. The most common AEs were colitis/diarrhea, rash, transaminitis, endocrinopathies, and other irAEs. SD was noted in four (12%) patients [45].

BBI608 in combination with pembrolizumab demonstrated antitumor activity with acceptable toxicities in a phase I/II study conducted by other investigators in 50 patients with metastatic colorectal cancer whose tumors had high microsatellite instability (Cohort A, *n* = 10) or were microsatellite stable (Cohort B, *n* = 40). The most common grade ≥3 treatment-related AEs were fever, anorexia, and diarrhea [46]. In Cohort A, the immune-related objective response rate (irORR) was 50.0% (CR, *n* = 1; PR, *n* = 4; 95% CI, 18.7–81.3) and 40% of patients had SD ≥ 3 months. The median PFS and OS was not reached. In Cohort B, the irORR was 10.0% (PR, *n* = 4; 95% CI, 2.8–23.7) and 17.5% of patients had SD ≥ 3 months. The median PFS was 1.6 months (95% CI, 1.4–2.1) and the median OS was 7.3 months (95% CI, 5.3–11.8 months) [46].

The strengths of our study include the investigation of novel combinations, close patient monitoring in a specialized Phase I unit, and timely management of TEAEs. The observation that selected patients had prolonged disease stabilization, progression-free survival, and overall survival is intriguing. Tumor molecular profiling (Appendix A) provided valuable biomarkers for the selection of innovative targeted therapies in a timely manner at the time of disease progression for some patients, which may have contributed to prolonged OS.

Our study has several limitations. First, it involves a small number of patients with various tumor types treated in a single institution. Non-compartmental PK analysis could not be performed because additional patients were treated at other institutions and their data are unavailable. Second, it lacks phospho-STAT3 expression assessment for patient selection. Third, correlative studies of biomarkers driven by STAT3 activity, signaling pathway abnormalities, and/or immunotherapy (e.g., PD-1/ PD-L1 expression on the tumor or tumor-infiltrating cells) were not performed. Furthermore, in our study, as in similar nonrandomized trials that combine novel agents with immunotherapy, it is impossible to determine whether BBI608 adds to the antitumor activity of the checkpoint inhibitors. Another limitation is that the use of subsequent therapy after progression may confound OS.

In conclusion, our study demonstrates that treatment with BBI608 combined with immunotherapy is feasible and overall well tolerated in this patient population treated in a specialized Phase I department. Encouraging antitumor activity was noted in selected patients who received BBI608 and nivolumab. Prospective studies to elucidate the tumor mechanism of response and resistance to BBI608 and checkpoint inhibitor combination therapy in selected tumor types and/or with molecular alterations are warranted. Innovative clinical trials investigating different combinations of targeted agents and immunotherapy that include molecular profiling and biomarker screening for optimization of treatment selection will accelerate the implementation of personalized medicine.

## Figures and Tables

**Figure 1 cancers-14-01330-f001:**
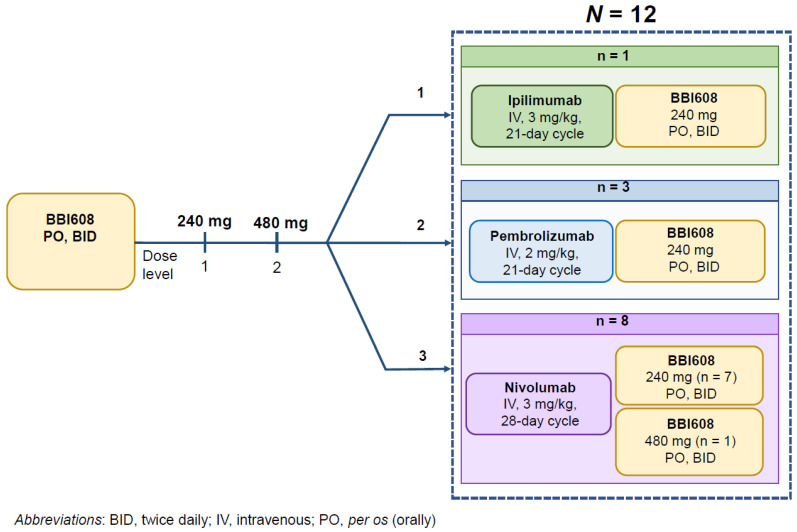
Study schema.

**Figure 2 cancers-14-01330-f002:**
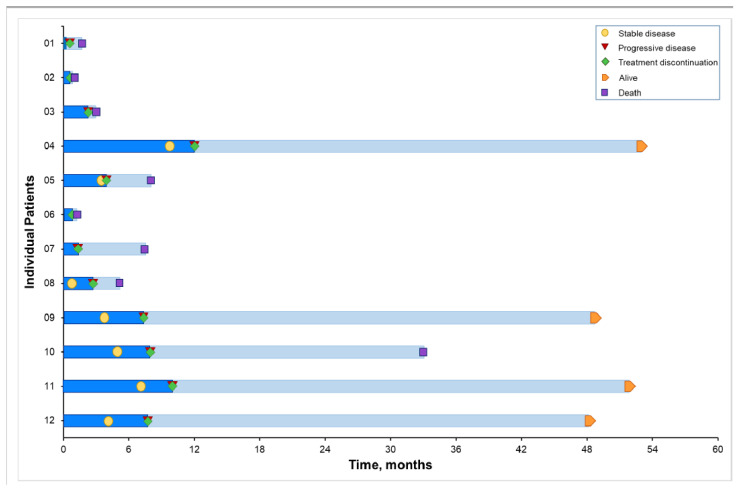
Swimmer plot. Clinical events in patients who underwent treatment with BBI608 and immunotherapy. The swimmer plot illustrates clinical responses in relationship to duration of treatment and time of treatment discontinuation. Cycle 1, day 1 was chosen as baseline (time 0). Symbols along and at the end of each bar represent relevant clinical events. Data cut-off, August 2021.

**Figure 3 cancers-14-01330-f003:**
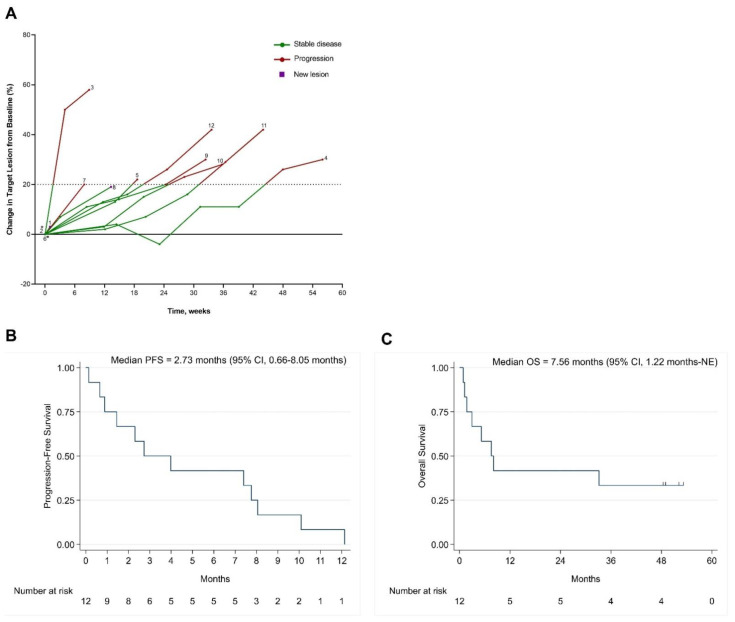
(**A**) Spider plot. Tumor growth or shrinkage from baseline in patients who underwent treatment with BBI608 and immunotherapy. Spider plot illustrates changes in lesion tumor burden and presence of new lesions over time. CT scan data at the most recent time prior to cycle 1, day 1 was chosen as baseline (time 0). The horizontal (x) axis shows time at restaging CT scans as weeks from baseline. The vertical (y) axis shows percentage of change in tumor measurement from baseline. Each line represents data from an individual patient and is labeled with the patient ID at the end. Each dot represents a data point collected at each restaging CT scan. The green lines represent stable disease. The red lines represent progressive disease (≥20% increase in tumor measurements from baseline), based on RECIST1.1. The square dot represents presence of new lesions. * Patients 02 and 06 discontinued treatment owing to toxicity before the completion of cycle 1 and did not have tumor assessment. (**B**) Kaplan–Meier curve for progression-free survival. (**C**) Kaplan–Meier curve of overall survival.

**Table 1 cancers-14-01330-t001:** Study design.

BBI608	BBI608, mg	Ipilimumab, 3 mg/kg	Nivolumab, 3 mg/kg	Pembrolizumab, 2 mg/kg
		(1C = 21D)	(1C = 28D)	(1C = 21D)
	Twice daily	D1	D1, 15	D1
Dose level				
1	240			
2	480			

Abbreviations: C, cycle; D, day.

**Table 2 cancers-14-01330-t002:** Baseline characteristics of 12 patients who underwent treatment with BBI608 and immunotherapy.

Pt. ID	Age, Yrs	Sex	Tumor Dx	ECOG PS	No. of Prior Rx	Prior Rx	No. of Metastatic sites	Metastatic Sites	Liver Metastasis	PLT, ×10^9^/L	LDH, U/L	Alb., g/dL	Cr, mg/dL	ALT/AST, U/L
01	Late 60s	F	Neuroendocrine carcinoma of the small bowel	1	2	Carboplatin-etoposide; topotecan	1	Liver	Yes	201	501	3.7	0.87	32/34
02	Late 70s	M	Mesothelioma of lung	1	1	Pemetrexed-carboplatin	1	Lung	No	219	383	3.3	0.95	36/23
03	Early 60s	M	Adenocarcinoma of the esophagus	1	4	Cisplatin-5-fluorouracil; docetaxel-oxaliplatin-5-fluorouracil; modified leucovorin calcium (calcium folinate)-5-fluorouracil-irinotecan; paclitaxel-ramucirumab; radiation	6	Left adrenal, peritoneum/retroperitoneum, pancreas, lymph nodes, abdominal wall musculature and bones, paraspinal	No	164	486	3.9	0.8	20/22
04	Early 50s	F	Adenoid cystic carcinoma of the Bartholin gland	1	3	Adjuvant pelvic radiation; vaginal brachytherapy; vulvar radiation.	6	Peritoneum, retroperitoneum, lung, lymph node, liver, spleen	No	225	409	4.1	0.62	46/33
05	Early 50s	M	Squamous cell carcinoma of the right anterior tongue and floor of the mouth	1	4	Docetaxel-carboplatin-5-fluorouracil; chemo-radiation with carboplatin; docetaxel, cisplatin/carboplatin +/− erlotinib; investigational pan-fibroblast growth factor receptor [FGFR] kinase inhibitor		Right perihilar mass	No	153	518	4	0.81	24/25
06	Early 60s	M	Adenocarcinoma of the pancreas	1	2	Gemcitabine-nab-paclitaxel; leucovorin calcium-fluorouracil-irinotecan hydrochloride-oxaliplatin	1	Liver	Yes	160	1007	3.9	0.69	35/17
07	Late 60s	F	Adenocarcinoma of the lung	1	3	Carboplatin-pemetrexed; pemetrexed-bevacizumab; pemetrexed-bevacizumab-carboplatin	3	Lung, pleural space, bone	No	153	663	4	0.91	27/33
08	Early 50s	M	Adenocarcinoma of the distal esophagus	1	4	Concurrent chemoradiation with 5-fluorouracil-docetaxel-cisplatin; folinic acid-5-fluorouracil-oxaliplatin; ramucirumab; 5-fluorouracil	3	Esophagus, lung, lymph nodes	No	279	385	4.1	0.96	79/86
09	Early 40s	F	Adenocarcinoma of the ovary	1	5	Folinic acid-fluorouracil-oxaliplatin; capecitabine-oxaliplatin-bevacizumab; investigational FGFR inhibitor; investigational micellar formulation of oxaliplatin; investigational pan-RAF inhibitor	1	Lung	No	350	354	4.9	0.57	38/30
10	Early 30s	M	Ex-pleomorphic adenoma of the right parotid	0	4	Adjuvant chemoradiation; cisplatin-proton radiation; carboplatin-docetaxel; investigational FGFR inhibitor	1	Lung	No	215	402	4.2	0.98	33/28
11	Late 30s	F	Adenoid cystic carcinoma of parotid gland	1	1	Surgery; adjuvant radiation therapy; investigational FGFR inhibitor	3	Lung, kidney, lymph nodes	No	255	409	4.1	0.69	40/25
12	Late 30s	F	Adenoid cystic carcinoma of the left parotid gland	1	0	Parotidectomy; postoperative radiotherapy; radiation therapy	5	Kidney, renal pelvis, liver, lung, bone	Yes	237	431	4.4	0.7	28/20

Abbreviations: Alb., albumin; ALT, alanine aminotransferase; AST, aspartate aminotransferase; Cr, creatinine; Dx, diagnosis; ECOG PS, Eastern Cooperative Oncology Group performance status; LDH, lactate dehydrogenase; PLT, platelet; Pt., patient; Rx, therapy. Reference ranges: albumin, 3.5–5.2 g/d; ALT, 0–41 U/L; AST, 0–40 U/L; creatinine, 0.51–0.95 mg/dL; LDH, 135–225 U/L; PLT, 140–440 × 109/L.

**Table 3 cancers-14-01330-t003:** Treatment-emergent adverse events (TEAEs) of patients treated with BBI608 and immunotherapy.

Adverse Event	All Grades		Grade 1		Grade 2		Grade 3		Grade 4	
(*N* = 12) *	No.	%	No.	%	No.	%	No.	%	No.	%
**Gastrointestinal**										
Diarrhea	5	41.7	3	25.0			2	16.7		
Nausea	4	33.3	3	25.0	1	8.3				
Abdominal pain	4	33.3	2	16.7	1	8.3	1	8.3		
Colitis	2	16.7					2	16.7		
Constipation	1	8.3	1	8.3						
**Musculoskeletal and connective tissue symptoms**										
Back pain	2	16.7	2	16.7						
Cancer-related pain	1	8.3	1	8.3						
Pain right side of face and jaw	1	8.3					1	8.3		
Trismus	1	8.3	1	8.3						
Leg swelling	1	8.3					1	8.3		
**Psychiatric symptoms**										
Anxiety	1	8.3	1	8.3						
**Skin and subcutaneous tissue symptoms**										
Dry skin/pruritus	2	16.7	2	16.7						
**Laboratory abnormalities**										
Elevated alkaline phosphatase	1	8.3	1	8.3						
Elevated ALT	1	8.3	1	8.3						
**General disorders and administration site conditions**										
Fatigue	3	25.0	1	8.3	1	8.3	1	8.3		
Gait instability	1	8.3					1	8.3		
General weakness	1	8.3	1	8.3						
**Metabolism and nutrition symptoms**										
Dehydration	1	8.3			1	8.3				
Hyperkalemia	1	8.3					1	8.3		
Hypokalemia	1	8.3	1	8.3						
Hyponatremia	1	8.3					1	8.3		
**Vascular symptoms**										
Hypertension	1	8.3	1	8.3						
**Endocrine symptoms**										
Hypothyroidism	2	16.7	1	8.3	1	8.3				
**Blood and lymphatic system symptoms**										
Anemia	2	16.7	1	8.3			1	8.3		
**Respiratory, thoracic, and mediastinal symptoms**										
Dyspnea	3	25.0					3	25.0		
Cough	2	16.7	2	16.7						
Hemoptysis	1	8.3					1	8.3		
Bronchitis	1	8.3					1	8.3		
Pulmonary embolism	2	16.7	1	8.3					1	8.3
**Infectious complications**										
Pneumonia	1	8.3					1	8.3		
Urinary tract infection	1	8.3	1	8.3						
**Others**										
Orange urine	1	8.3	1	8.3						

* Patients are counted only once per adverse event and severity classification (the most severe adverse event is shown). Abbreviations: ALT, alanine aminotransferase.

**Table 4 cancers-14-01330-t004:** Clinical outcomes of 12 patients who underwent treatment with BBI608 and immunotherapy.

Pt. ID	Cohort	Treatment Arm	BBI608 Dose Level, mg, PO BID	Tumor Type	No. of Cycles	Best RECIST Response	PFS *, Months	Progression Status	Subsequent Therapy	Survival Status	OS ^†^, Months
01	1	Ipilimumab	240	Neuroendocrine carcinoma of the small bowel	<1 ^‡^	PD	0.1	Yes	None	Deceased	1.7
02	2	Pembrolizumab	240	Mesothelioma of lung	<1 ^§^	PD	0.7	No	None	Deceased	0.9
03	2	Pembrolizumab	240	Adenocarcinoma of the esophagus	2	PD	2.3	Yes	None	Deceased	3.0
04	3	Nivolumab	240	Adenoid cystic carcinoma of the Bartholin gland	13	SD	12.1	Yes	Investigational therapy: bevacizumab-temsirolimus-valproic acid; Anti-Globo H mAb; HDAC6 inhibitor	Alive	53.0
05	3	Nivolumab	240	Squamous cell carcinoma of the right anterior tongue and floor of the mouth	4	SD	4.0	Yes	None	Deceased	8.1
06	3	Nivolumab	240	Adenocarcinoma of the pancreas	<1 ^‖^	Clinical progression	0.9	No	None	Deceased	1.2
07	3	Nivolumab	240	Adenocarcinoma of the lung	2	PD	1.4	Yes	Radiation therapy; poziotinib	Deceased	7.5
08	2	Pembrolizumab	240	Adenocarcinoma of the distal esophagus	3	PD	2.7	Yes	None	Deceased	5.2
09	3	Nivolumab	240	Adenocarcinoma of the ovary	7	SD	7.4	Yes	Radiation therapy; palbociclib	Alive	48.7
10	3	Nivolumab	240	Ex-pleomorphic adenoma of the right parotid	9	SD	8.0	Yes	Investigational therapy: FGFR inhibitor; nivolumab/ipilimumab; capecitabine	Deceased	33.0
11	3	Nivolumab	480	Adenoid cystic carcinoma of parotid gland	11	SD	10.1	Yes	Investigational therapy: HDAC-6 inhibitor; MoAb Globo H inhibitor; PI3K inhibitor-nivolumab; nitro-benzene-aldo-keto reductase 1C3-activated prodrug; fludarabine-cyclophosphamide-T-cell therapy; MoAbs targeting LAG3 and TIM3; radiation therapy	Alive	51.9
12	3	Nivolumab	240	Adenoid cystic carcinoma of the left parotid gland	8	SD	7.7	Yes	ERK 1/2 inhibitor; radiation therapy; lenvatinib	Alive	48.2

Abbreviations: BID, twice daily; N/A, non-applicable; OS, overall survival; PD, progressive disease; PFS, progression-free survival; PO, per os (orally); SD, stable disease. * Progression-free survival was measured in months from cycle 1, day 1 to date of disease progression or treatment discontinuation. ^†^ Overall survival was measured in months from cycle 1, day 1 to time of death or last follow-up. ^‡^ Patient 01 received one dose of ipilimumab and three days of BBI608 (240 mg BID). She was taken off protocol owing to new brain metastasis before the completion of cycle 1. ^§^ Patient 02 received one dose of pembrolizumab. He was hospitalized and stopped taking BBI608 after being discharged. The patient died before the completion of cycle 1 and did not have tumor assessment. ^‖^ Patient 06 received one dose of nivolumab and discontinued the study treatment before the completion of cycle 1 owing to failure to thrive. He did not have tumor assessment.

## Data Availability

Data are available upon reasonable request. The datasets used and/or analyzed during the current study are available from the corresponding author upon reasonable request and approval from study sponsor and institution according to available guidelines at the time of request.

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
