# Peer review of "Ipilimumab, Pembrolizumab, or Nivolumab in Combination with BBI608 in Patients with Advanced Cancers Treated at MD Anderson Cancer Center"

_cancers, 2022, doi:10.3390/cancers14051330_

Round 1
Reviewer 1 Report
This paper describes the co-treatment of twelve patients with advanced solid tumours with one of three immune checkpoint inhibitors and the small molecule inhibitor of cancer stemness pathways BBI608. BBI608 is a novel oral first-in-class cancer stemness inhibitor. The rationale of the study was that targeting putative cancer stem cells (CSCs) would ameliorate resistance to checkpoint inhibitors. The study aimed to determine safety and efficacy of treating with BBI608 alongside standard therapy. There have been promising pre-clinical data showing the efficacy of targeting CSCs by disrupting stemness pathways in combination with standard therapies.
There have been few clinical trials of BBI608 to date but those that have been published report low toxicity and some suggestion of anti-tumour activity. The paper under review adds to the growing evidence for the safety of this orally available drug in patients with advanced disease.
In discussion of the limitations of the study the authors observe that the use of subsequent therapy after progression may confound the observed overall survival. Additionally, the authors state "it is impossible to determine whether BBI608 adds to the antitumor activity of the checkpoint inhibitors ." Therefore, the title of the paper and lines 35-36 in the abstract “Several patients had prolonged disease stabilization and overall survival” should be reworded to avoid misinterpretation that OS can be attributed to BBI608.
The tables and figures are clear and aid understanding of the data.
Could the authors please comment on the PK profile for patient 012, which only has data points early in the time course (Figure S3. BBI608-201CIT Individual PK Profiles). As patient 012 achieved prolonged overall survival, it is of interest if the plasma levels of BI608 was below detectable levels.
Reviewer 2 Report
In this manuscript entitled Safety and efficacy of BBI608 in combination with ipilimumab, pembrolizumab, or nivolumab in patients with advanced cancers treated at MD Anderson Cancer Center et al, the authors discuss the use of ROS generating agent in immunotherapy-based combinations.
1. In the introduction, the validity of the combination being examined is not properly explained. The authors themselves state that the potential action is theoretical and not supported by experimental data. Moreover, ROS generation has already been shown to be toxic to cancer cells but also to T lymphocytes. Extensive ROS production disrupts lymphocyte functions such as cytotoxic activity. Therefore, the rationale for combining the proposed therapies is unclear.
2. The number of patients in this research is very low. In addition, the lack of comparison with a single treatment and rdomisation does not make it possible to evaluate the effectiveness. Therefore, the title should be changed.
3. The description of the results should be more precise and less speculative. For example, part of the PFS needs to be rewritten
4. Less important changes concern shortcuts - all abbreviations should be explained except the obvious ones like DNA etc.
Reviewer 3 Report
The present study reports on the results of a phase 1 study evaluating an investigational reactive oxygen species generator compound in combination with different immune checkpoint inhibitors.
- The authors should provide additional details on the methodologies used for tumor molecular profiling
- A table summarizing the efficacy outcomes in the three cohorts would be useful.
- A summary of statistics for PK parameters would be useful.
Round 2
Reviewer 2 Report
The manuscript can be published in a present form.